behaviour, evolution

optimal group size, fitness, folivore paradox, infanticide, reproduction, survival

**Authors for correspondence:**
Elizabeth Tinsley Johnson
e-mail: etinsley@msu.edu
Noah Snyder-Mackler
e-mail: nsnyderm@asu.edu
Jacinta C. Beehner
e-mail: jbeehner@umich.edu

†Co-first authors.
‡Co-senior authors.

# The Goldilocks effect: female geladas in mid-sized groups have higher fitness

Elizabeth Tinsley Johnson[1,†], Jacob A. Feder[2,†], Thore J. Bergman[4,5], Amy Lu[2,3], Noah Snyder-Mackler[6,7,‡] and Jacinta C. Beehner[4,8,‡]

[1]Department of Integrative Biology, Michigan State University, East Lansing, MI 48824, USA
[2]Interdepartmental Program in Anthropological Sciences, and [3]Department of Anthropology, Stony Brook University, Stony Brook, NY 11794-4364, USA
[4]Department of Psychology, University of Michigan, Ann Arbor, MI 48109-1043, USA
[5]Department of Ecology and Evolution, University of Michigan, Ann Arbor, MI 48019-1085, USA
[6]School of Life Sciences, Arizona State University, Tempe, AZ 85287-4701, USA
[7]Center for Evolution and Medicine, Arizona State University, Tempe, AZ 85287-1701, USA
[8]Department of Anthropology, University of Michigan, Ann Arbor, MI 48109-1107, USA

ETJ, 0000-0002-1291-1261; JAF, 0000-0002-3087-7298; TJB, 0000-0002-9615-5001; AL, 0000-0003-4758-216X; NS-M, 0000-0003-3026-6160; JCB, 0000-0001-6566-6872

The cost–benefit ratio of group living is thought to vary with group size: individuals in 'optimally sized' groups should have higher fitness than individuals in groups that are either too large or too small. However, the relationship between group size and individual fitness has been difficult to establish for long-lived species where the number of groups studied is typically quite low. Here, we present evidence for optimal group size that maximizes female fitness in a population of geladas (*Theropithecus gelada*). Drawing on 14 years of demographic data, we found that females in small groups experienced the highest death rates, while females in mid-sized groups exhibited the highest reproductive performance. This group size effect on female reproductive performance was largely explained by variation in infant mortality (and, in particular, by infanticide from immigrant males) but not by variation in reproductive rates. Taken together, females in mid-sized groups are projected to attain optimal fitness due to conspecific infanticide and, potentially, predation. Our findings provide insight into how and why group size shapes fitness in long-lived species.

## 1. Introduction

Variability in group size within species reflects a balance between the costs and benefits of group living [1,2]. For example, individuals living in large groups may experience high within-group foraging competition (a cost; e.g. [3–5]) but low predation risk (a benefit; e.g. [3,6–9]), or high parasite loads (a cost; e.g. [10]) but improved resource defence (a benefit; e.g. [11]). Individuals living in optimally sized groups, where (by definition) the benefits outweigh the costs, should have the highest lifetime reproductive success (i.e. fitness [12]) compared to others in the population [13]. Although the link between group size and fitness has long been the focus of socioecological theory, it has been challenging to demonstrate this relationship with empirical data. This is, in large part, because group size fluctuates over an individual's lifespan, making it difficult to align a specific group size with a clear fitness metric (especially in long-lived species). As a result, testing the effects of group size requires long-term data across multiple groups of variable size.

Ecological costs and benefits at both extremes of group size support the idea that mid-sized groups are optimal. For example, the feeding costs of living in large groups are well documented: variables that indicate greater within-group feeding competition, such as home range size, day range length and feeding time are often positively correlated with group size (carnivores [14];

primates [15,16]), and those that reflect the consequences of such competition, such as body condition, fertility and survival are often negatively correlated with group size [17–22]. Despite the costs of living in large groups, individuals are often found in groups significantly larger than the theoretical species optimum [23,24], probably because splitting into two smaller groups is equally, if not more, costly. This may be particularly true if small groups are more susceptible to predation [25,26] or if between-group competition pushes competitively weaker groups into inferior home ranges (carnivores [27]; primates [28,29]). Perhaps as a consequence of costs at both extremes, individuals in both large and small groups exhibit higher glucocorticoid levels, greater ranging disadvantages, and experience more extreme energetic conditions when compared to individuals in mid-sized groups (e.g. [16,30–32]; reviewed in [5]).

However, many species do not exhibit group sizes that conform to predictions based on ecological factors (e.g. [33]). For example, although folivores are thought to experience low within-group feeding competition, females in numerous folivorous taxa live in groups that are smaller than those found in frugivorous species (the 'folivore paradox' [34]; e.g. [35]). For some folivorous species, within-group feeding competition may still be high, thus limiting group size (primates [36]; elephants [37]; deer [38]). Alternatively (or additionally), in many folivorous primates, females may form smaller groups because these groups are less attractive to immigrant males and thus less vulnerable to infanticide [39,40]. By contrast, in some species, infanticide risk may favour larger groups that are better able to prevent male immigration or defend dependent offspring [41]. As a result, the conspecific threat of infanticide has the potential to shape group size such that mid-sized groups are optimal, but isolating the effects of such social factors from environmental factors has proven challenging.

Geladas are folivorous monkeys that feed on widely dispersed food resources consisting primarily of graminoid (grass and sedge) leaves [42,43]. Due to this ubiquitous resource base, geladas face low within-group feeding competition [44] and (as predicted [34]) live in large groups for predator protection [45,46]. However, gelada social organization is not so simple: individuals form small reproductive groups (hereafter, 'units', varying in size from 1 to 12 adult females [47]) which, in turn, unite to form large bands (100+ individuals) that travel and forage together in the same home range (forming a 'multi-level society' [46]). The large foraging band likely represents the ecological unit (i.e. by offering some degree of protection from predation [48]) and conforms to the predictions of standard socioecological theory [34]. By contrast, the small reproductive unit represents the social unit and, as such, is likely shaped by social factors. Specifically, male 'takeovers', where a new immigrant male replaces the dominant male, have the potential to dramatically alter female reproductive patterns via sexual conflict—infanticide and the Bruce effect [49,50]. Infanticide is the leading cause of infant mortality: immigrant males can kill up to half of the dependent infants in a unit [51]. Moreover, larger units experience more takeovers than smaller units [52].

Ecological factors like habitat quality are unlikely to vary significantly among units in the same band; therefore, we predict that females in larger units incur disproportionately higher fitness costs due to male takeovers compared to females in smaller units. Alternatively (or additionally), when a takeover occurs, infants in small groups may have a higher chance of being killed due to poor infant defence. If both large and small units incur fitness costs due to infanticide, then we predict that females in mid-sized units will exhibit the highest fitness. Finally, adult female survival may also be affected by unit size: females in small units may be more vulnerable to predation (e.g. [25]), while females in larger units may be more vulnerable to disease (e.g. [53]). If both factors affect female longevity in geladas, then we predict the lowest adult female death rates in mid-sized units.

To test these predictions, we first examined the evidence for an optimal unit size for gelada females. Specifically, we tested if unit size predicted female death rates and/or reproductive performance (the production of surviving infants). Second, because reproductive performance could be a product of both female fertility and/or infant mortality, we explored the effect of unit size on these two components. Specifically, we assessed variation in both interbirth intervals (IBIs), which should track both female energetic condition (e.g. [54,55]; reviewed in [7]) and takeover-related fetal loss [49,56], and in infant mortality, which is predominantly driven by infanticide for this population [51]. Third, we examined how the cause of infant deaths varied by unit size, specifically considering the extent to which maternal death and/or infanticide explained these patterns. Taken together, our results show that females in mid-sized units appear to display optimal fitness, resulting from the combined effects of lower adult female death rates and infanticides in these units.

## 2. Material and methods

### (a) Study site and subjects

The data for this study derive from 14 years of observation (Jan 2006–Jul 2020) on a population of wild geladas living in the Simien Mountains National Park, in northern Ethiopia (13°13.5′ N latitude). The Simien Mountains Gelada Research Project (SMGRP, formerly the University of Michigan Gelada Research Project) has collected near-daily behavioural, demographic, genetic and hormonal data from individuals since Jan 2006. All gelada subjects are habituated to human observers on foot and are individually recognizable. We used longitudinal data from 200 adult females in 46 reproductive units (20 original 'founding' units that later formed 26 units following fissions).

### (b) Unit size

The identities of all individuals in a unit were recorded each day the unit was seen. For each month of the study period, we recorded the total number of adult females in each unit; where changes in unit size occurred (i.e. due to adult female deaths or subadult female maturations), we used the maximum number of adult females in a unit in a given month. We focus on adult females because we have longitudinal records of the number of adults in each unit for the entire study period but only started recording the total number of individuals (including juveniles and infants) in 2012. However, when we compared the number of adult females in a unit to the total number of individuals in a unit for the subset of data where we have both (2012–2020), we found that both values were highly correlated (Pearson correlation coefficient = 0.76, $p$-value = $2.2 \times 10^{-16}$).

Gelada units contain one reproductively dominant 'leader male' as well as 0–3 'follower males' that receive few reproductive opportunities but can deter male takeovers [47]. Because the number of females and the number of males could have contrasting impacts on female fitness components, and because the number of males was weakly correlated with the number of females (Pearson correlation coefficient = 0.21, p-value = 2.2 × $10^{-16}$), we did not include males when calculating group size. In all relevant analyses, we considered the number of males separately as covariates.

Changes in the number of adult females within a unit were primarily due to female maturations and deaths. Maturations were recorded as the first observation of a sex skin swelling (details are outlined in [57]). Deaths were recorded as the first day an adult female was no longer observed with a unit for more than three consecutive encounters with that unit (and not observed in a different unit, as in the case of transfers or fissions).

Dates of fissions, fusions and female transfers were recorded as the first day the unit females were no longer observed together and subsequently observed either in a separate unit with a new leader male (for fissions), together with non-unit females and a new leader male (for fusions), or associating individually with non-unit females and a new leader male (for the rare cases of female dispersal). In all cases, we immediately identified known females in fissioned units or in new units following their disappearance from their natal unit.

All models included female unit size as a continuous predictor variable (see electronic supplementary material, figure S1 for the observed distribution of unit sizes). However, for visualization purposes in our figures, we also categorized units into small, medium (i.e. mid-sized) and large units based on the observed range in unit sizes. Specifically, cut-offs were determined by calculating the tertiles of the observed monthly distribution of sizes: 'small' indicates units of less than or equal to 4 adult females; 'mid-sized' indicates units from 5–7 adult females; and 'large' indicates units of 8 or more adult females.

## (c) Adult female death rates

For this analysis, we included data on 200 adult female geladas. All adult females had known or estimated birth dates from which we calculated age. Dates of birth were known for 56 adult females born during the study, while the remaining females' birth dates were estimated using their size at the start of observation (n = 44), back-calculated using life-history milestones (i.e. maturation or first birth, n = 40), or extrapolated from their total number of infants (n = 60). The mean age at the mid-point of the study (2013) for all females was 11.86 ± 5.14 s.d. years (range = 3.54–28.90 years). There was no mean difference in female age across unit sizes (estimated or known; females estimate = −2.01 ± 2.35 s.e., p-value = 0.394), but large units skewed towards having more young females than small (Kolmogorov–Smirnov D = 0.094, p-value = 0.049) and midsized units (D = 0.104, p-value = 0.012; electronic supplementary material, figure S2).

To assess whether the adult female death rate varied by unit size, we constructed a binomial generalized linear mixed model (GLMM) using the lme4 package (v. 1.1.20 [58]) in R (v. 3.6.0 [59]). Here, the unit of observation was the female-year (1 Jan–31 Dec, unless truncated by maturation, death or the start or end of the study period). The dependent binary variable indicated whether the female died during the study year, which was modelled as a function of the following predictors: unit size (the average number of females in the unit, including both the linear and the quadratic term), the female's age at the start of the year (linear and quadratic) and the number of males (i.e. the total number of adult follower males plus one leader male). We controlled for the repeated measures of female, year and

unit as crossed random effects (due to unit fissions, females can reside in multiple units during their lifetimes). Additionally, we constructed mixed-effects Cox proportional hazards models on adult female survival using the R package coxme [60]. For these models, we incorporated uncertainty in female ages by sampling randomly from females' minimum and maximum birth dates. Model averaged results from 1000 survival model simulations were congruent with the results from the GLMM (electronic supplementary material, table S6).

## (d) Reproductive performance

For the majority of infants born during the study period (n = 352 out of 394 total), the date of birth was known within a day or two. For those infants where the exact date of birth was unavailable (n = 42), we were able to assign a birth date within one month of the actual birth date based on established morphological and behavioural criteria (i.e. the size of the infant, infant motor skills and the presence/absence of the umbilical cord; for more details; see [57]). Conception dates were retrospectively assigned by subtracting the mean gestation length (n = 183 days; see [57]) in the population from the observed or estimated date of birth.

We used a binary scoring system to assign female reproductive performance on a monthly basis. Females were assigned a 1 in a given month if they conceived an infant that survived to 1.5 years of age (the mean age of weaning in this population [57]), regardless of actual weaning date. Females were assigned a 0 if they conceived an offspring that died before reaching 1.5 years of age or if they did not conceive during the calendar month. We excluded all months and births after 2018, as the survival outcomes of these infants were not resolved by the end of the study period.

To assess the effects of unit size on female reproductive performance, we constructed a binomial GLMM. The dependent binary variable indicated whether a female conceived a surviving offspring during each female-month (273 surviving offspring, 996.5 female-years total, 188 females). We modelled this outcome variable as a function of the following predictors: female age (both the linear and quadratic term, to control for the known effects of female age on reproductive output, reviewed in [61]), number of females and the number of males. We controlled for the repeated measures of individual identity, unit and year as crossed random effects. To capture the effect of unit size variation beyond the month of conception, we additionally modelled reproductive performance using 6-month bins (i.e. Jan–Jun, Jul–Dec) that were centred on the two primary conception seasons [50]. We used the same binary outcome listed above and included the average number of females and males as fixed effects. The results from this separate analysis (electronic supplementary material, table S7) were congruent with those generated from the model using monthly data.

Given the long lifespans of geladas, we do not have the data necessary to directly calculate lifetime reproductive success. However, we estimated this value by pairing median ages at death with reproductive rates across the three unit size categories (i.e. small, medium and large). Specifically, we first calculated median ages at death for adult females at each unit size category from survival curves including right-censored females (electronic supplementary material, figure S3). Second, we calculated reproductive rates by summing the total number of surviving offspring within each unit size category and then dividing by the total number of female-years. To approximate lifetime reproductive success, we subtracted the population mean age at first birth (6.06 years; range: 4.88–7.56 [57]) from the median ages at death and multiplied these reproductive lifespans by their corresponding reproductive rates. Although these static estimates may not reflect true differences in lifetime reproductive success

Proc. R. Soc. B 288: 20210820

 *Proc. R. Soc. B* **288**: 20210820

(because unit size changes across the lifespans of individual females, especially following fissions and fusions), we use these estimated fitness differences to underscore the potential costs of remaining at suboptimal unit sizes and to highlight the need for female strategies to avoid these costs.

### (e) Interbirth intervals

Using the birth dates of all infants in this population, we calculated IBIs between females' successive offspring. We constructed two linear mixed-models (LMM) using the *lme4* package. For both models, the dependent variable was the length of the IBI in days, which was modelled as a function of unit size at the first infant's birth (both linear and quadratic), the mother's age, and the infant's sex. In the first model, we used only IBIs following surviving offspring ($n = 187$), as infant mortality substantially shortens IBIs in this population and elsewhere [51]. However, because IBIs where takeovers occur may capture instances of takeover-related fetal loss (i.e. the 'Bruce effect' [49]), which can lengthen IBIs [56], in the second model we only included the subset of IBIs that did not contain takeovers or infant deaths ($n = 82$). Rather than reflecting fetal loss, variation in these 'takeover-independent' IBIs should reflect the ecological advantages (e.g. higher food intake) that some females have over others in their ability to wean offspring and conceive again.

### (f) Infant mortality

The disappearance of any infant prior to the average age at weaning in this population (1.5 years of age [57]) was assumed to be a case of infant mortality. To assess infant mortality and its causes across unit sizes, we constructed a mixed-effects Cox proportional hazards model using the R package *coxme* [60]. For this model, we used all 394 infants born during the study period. Infants entered the dataset at birth and were censored at death ($n = 90$) or if their social units stopped being observed ($n = 17$) and exited the dataset when they survived to 1.5 years of age ($n = 287$). We used the number of females present during the month of the infant's birth as a fixed effect, including both linear and quadratic terms. We included birth year, unit and maternal ID as random effects. Schoenfeld residuals showed no significant deviations from the assumption of proportional hazards.

### (g) Takeover rates

We recorded the dates of all observed male takeovers ($n = 80$) of known reproductive units (following [50]) as well as the number of mature females in the unit at the time of takeover. We calculated the length of time each unit was seen at a given size for each year the unit was observed (244 total unit-years) and counted the number of takeovers that occurred during these periods. To calculate the influence of unit size (i.e. a number of adult females) on the likelihood of takeover, we modelled the number of takeovers a unit experienced as the dependent variable in a Poisson GLMM. Since takeovers are more likely to occur during longer observation windows, we included the number of continuous months the unit was seen at a given size as an offset term. To control for repeated measures across units and time, we included unit and year as random intercepts. Finally, we included two fixed-effect variables: the number of adult females in the unit and the average number of males.

### (h) Causes of infant mortality

To examine whether the probable causes of infant mortality varied with unit size, we assigned the cause of mortality based on the following characteristics: if the infant's death occurred within six months of a takeover, the cause of death was recorded as 'infanticide' ($n = 31$) [50,51]. If the infant died within three months of their mother's death or disappearance, the cause of death was recorded as 'maternal death' ($n = 27$). If maternal and infant disappearances co-occurred within the six months following a male takeover; however, the cause of death was categorized as 'maternal death' ($n = 4$). All other causes of infant deaths were recorded as 'unknown' ($n = 32$). For each of these mortality outcomes, we constructed a binomial GLMM using the *lme4* package. Here, we modelled whether the infant died of each respective mortality cause (i.e. maternal death, infanticide and unknown) as a binary variable, using unit size (both linear and quadratic) as a fixed effect and unit as a random effect. For these three analyses, we used only infants whose survival outcomes were known ($n = 377$; i.e. no right-censored offspring). All figures were constructed using *ggplot2* [62], and prediction intervals were extracted from mixed-models using the *effects* package [63].

## 3. Results

### (a) Adult female death rates

Of the 200 adult females included, 90 died before the end of the study period (median age at death = 16.2 years, including right-censored females). Unsurprisingly, the odds of dying increased substantially with age (age estimate = $30.07 \pm 4.06$ s.e., *p*-value = $2.1 \times 10^{-5}$). Females in small units had the highest death rates (females estimate = $-9.39 \pm 4.78$ s.e., *p*-value = 0.049; figure 1*a*; electronic supplementary material, table S1). Specifically, females in small units had an annual death rate of 10.4%, while females in medium and large units had an annual death rate of 5.8% and 7.3%, respectively.

### (b) Reproductive performance

Although reproductive units in this population can range in size from 1 to 12 adult females, females in units in the middle of this range (i.e. between 5 and 7 adult females, hereafter 'mid-sized' units) had the highest reproductive performance. Females in mid-sized units were most likely to conceive surviving offspring (females$^2$ estimate = $-18.14 \pm 7.69$ s.e., *p*-value = 0.018; figure 1*b*; electronic supplementary material, table S1) and were 17.6% more likely than females in small units and 39.8% more likely than females in large units to produce a surviving offspring. Given the median ages at death calculated from survival curves (16.0 years in small units, 19.7 years in mid-sized units, and 16.0 years in large units; electronic supplementary material, figure S3), females are projected to produce approximately 4.35 surviving infants in mid-sized units, compared to 2.70 infants in small units, and 2.27 infants in large units.

### (c) Interbirth intervals

IBIs tended to be slightly longer in larger units (females estimate = $492.6 \pm 242.6$ s.e., *p*-value = 0.044; females$^2$ estimate = $388.9 \pm 226.4$ s.e., *p*-value = 0.088; figure 2*a*). However, this effect was no longer significant when only IBIs that were uninterrupted by takeovers were included for analysis (females estimate = $39.2 \pm 230.3$ s.e., *p*-value = 0.867; females$^2$ estimate = $376.8 \pm 224.8$ s.e., *p*-value = 0.107; figure 2*b*; electronic supplementary material, table S2).

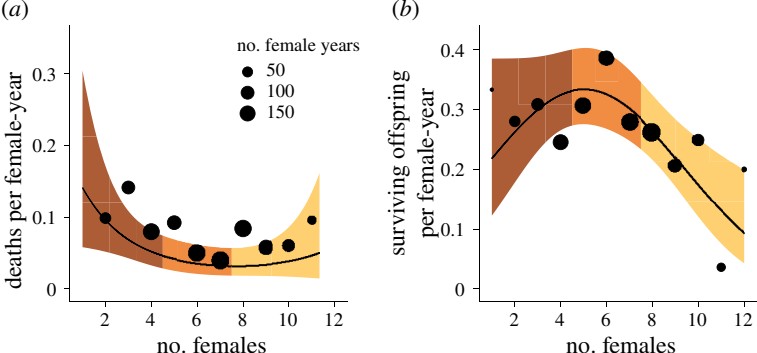

**Figure 1.** Female fitness is optimal in mid-sized units. (*a*) Females in mid-sized units had the lowest annual probability of death, calculated as the number of deaths observed at each unit size divided by the total number of female-years. Confidence bands show the lower and upper limits of the model predictions, with colours following the unit size categories described in *Methods*. The size of each point is proportional to the number of female-years observed at each unit size. (*b*) Females in mid-sized units showed the highest rates of conceiving surviving offspring. Points indicate yearly rates of conceiving surviving offspring at each unit size, generated from raw data. The size of each point is proportional to the number of female-years observed at each unit size. Confidence bands show the lower and upper limits of the model predictions. Although the model output specified monthly probabilities, we converted these to annual probabilities for the sake of visualization. (Online version in colour.)

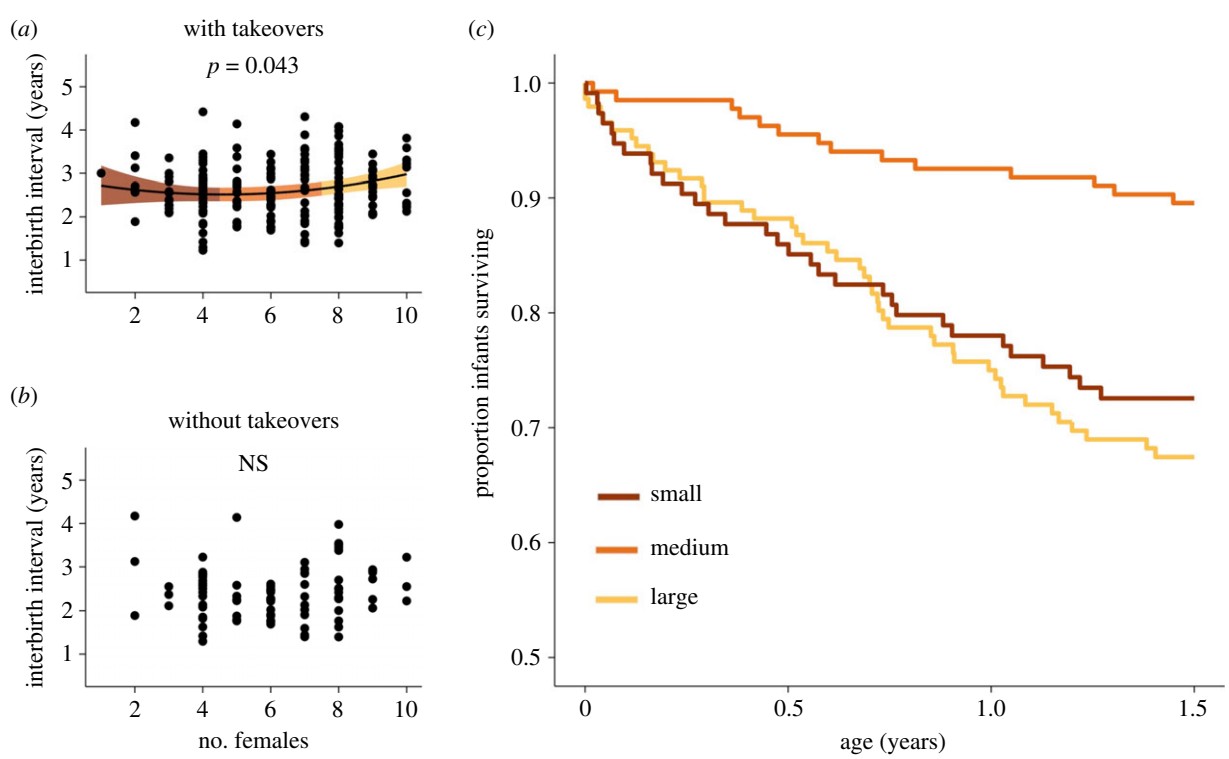

**Figure 2.** Unit size does not influence takeover-independent IBIs, but does impact infant survival. (*a*) Including intervals where takeovers occurred, females in large units show slightly longer IBIs. Coloured bands show the 95% confidence interval from the LMM. (*b*) Excluding cases where takeovers occurred, there is no relationship between unit size and IBI. (*c*) Infant survival was much higher in mid-sized units than in small and large units. Unit size was modelled as a continuous variable, and coloured categories here were used for visualization purposes only. (Online version in colour.)

### (d) Infant mortality

Out of 394 infants, 90 (22.8%) died before reaching 1.5 years. Infants born into small and large units were more likely to die before this age than infants born into medium-sized units (HR = 1.06 ± 0.017 s.e., *p*-value = 0.0014; figure 2*c*; electronic supplementary material, table S3). Specifically, 28.2% and 33.8% of infants born into small and large units died before weaning, while only 10.4% of infants born into mid-sized units died before reaching this age.

### (e) Takeover rates

Takeover frequency increased with unit size (females estimate = 7.50 ± 2.47 s.e., *p*-value = 0.003; figure 3*a*; electronic

supplementary material, table S4). On average, small units experienced male takeovers once every 4.15 years, mid-sized units once every 3.17 years, and large units once every 1.61 years. However, the number of males did not influence the frequency of takeovers (males estimate = −0.13 ± 0.13 s.e., *p*-value = 0.308; electronic supplementary material, table S4).

### (f) Causes of infant mortality

Infanticides and maternal death accounted for 38.9% and 25.6% of all dependent infant deaths, respectively. Despite the fact that small units experienced the lowest takeover rates, females in mid-sized units experienced the

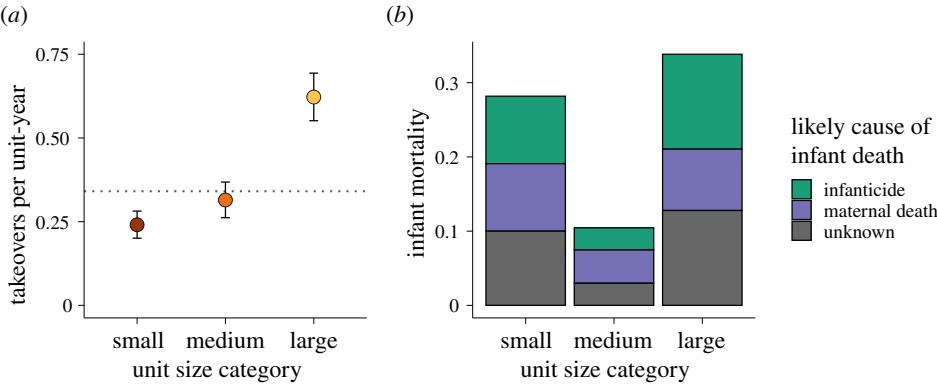

**Figure 3.** Takeovers and the causes of infant mortality vary with unit size. (*a*) Annual takeover rates increased with unit size. The dotted line represents the population mean (0.341 takeovers per unit-year). (*b*) Infants in small and large units were more likely to die of infanticide, but not maternal death or unknown causes. For the plots above, unit size categories are used solely for visualization purposes. (Online version in colour.)

lowest infanticide rates (females$^2$ estimate = 10.81 ± 3.72 s.e., $p$-value = 0.004; figure 3*b*; electronic supplementary material, table S5): 9.1% and 12.8% of infants born into small and large units experienced infanticide, respectively, while only 3.0% of infants in mid-sized units experienced infanticide.

Second, infants in mid-sized units were about half as likely to die as a result of maternal death as females in small and large units (females$^2$ estimate = 5.25 ± 3.38 s.e., $p$-value = 0.12, figure 3*b*; electronic supplementary material, table S5), although this effect was not significant. Specifically, 9.1% and 8.3% of infants born into small and large units died following their mothers' deaths, while only 4.5% of infants in mid-sized groups died under these circumstances. Lastly, there was no association between unit size and infant mortalities of unknown cause.

## 4. Discussion

Taken together, our results show that females in mid-sized units have the highest fitness. Females in smaller units die at the highest rates and females in both smaller and larger units have lower reproductive performance, primarily as a result of takeover-related infanticide. Specifically, male takeovers are more frequent in larger units and are more likely to lead to infanticide when they do occur in smaller units. We also show that infant death rates, but not takeover-independent female fertility (i.e. uninterrupted IBIs), vary by unit size, with the lowest infant death rates occurring in mid-sized units. As a result, females in mid-sized units demonstrate the highest reproductive performance and, when factoring in variation in adult female death rates, are expected to produce more offspring over the course of their reproductive lifespans than are females in small or large units.

Infants in small and large units were more likely to die from infanticide than infants in mid-sized units. The conspecific threat of infanticide has been suggested to limit group sizes in folivores (i.e. the 'folivore paradox' [39]). However, larger groups have also been suggested to act as a deterrent to male takeover and infanticide [41]. Here, we show that both forces may work in conjunction to result in optimal mid-sized groups. Infanticide in large units was driven by high rates of male takeovers. By contrast, we are less certain why infanticide in small units was also high, despite infrequent takeovers. One possibility is that small units have

poor infant defence (e.g. as suggested in grey langurs [41]). Another possibility is that new males may be more likely to commit infanticide in smaller units because smaller units present fewer immediate reproductive opportunities (i.e. receptive, cycling females) than larger units. A third possibility is that infanticide risk may follow a 'dilution effect' (similar to predation risk [64]), where infants in large units are buffered by an increased number of potential targets.

Our proxy for ecological factors, IBI [7], did not explain variation in reproductive performance across unit sizes (after removing cases where male-mediated fetal loss may have occurred). This is not surprising given the relatively dispersed and non-monopolizable diet of geladas. This analysis, however, did not consider whether unit size interacts with other metrics that might track ecological advantages, such as reproductive ageing, age at maturation or age at first birth. Based on a recent analysis on female maturations for this population [65], we expect that females in small units will have later ages at maturation and first birth to avoid inbreeding (since their fathers have longer tenures), and we expect that females in large units will have earlier ages at maturation and first birth due to the 'Vandenbergh effect' that accompanies male takeovers. However, even if these predictions are supported in future analyses, we do not expect these unit size differences to significantly alter our reproductive success estimates for two reasons: first, the observed range in age at first birth in this population is narrow [57], and second, in similarly long-lived primates, age at first birth had a very modest impact on variation in lifetime fitness [66,67]. Future analyses that address the potential for different female reproductive strategies, which require more complete female lifespans than we have currently, will allow us to determine the effects of unit size on realized lifetime reproductive success.

Adult female death, which contributes to low lifetime reproductive success, was highest in small units. Although female deaths could be triggered by takeover-related injuries [56], smaller units experienced less frequent takeovers than mid-sized or large units, and we did not observe an increase in adult female deaths after takeovers (in contrast with infant deaths, which do increase following takeovers [56]). Females in small units may be more likely to die if females in larger units monopolize food resources. Indeed, strong between-group competition has been shown in other folivorous species (e.g. [68,69]); however, in geladas, feeding-related aggression between units is relatively rare and low-intensity

[70]. Moreover, if between-group competition was strong, we would expect females in the smallest units to show the longest IBIs. We think the most likely explanation is that females in small units are more vulnerable to predation. Although the large foraging bands characteristic of gelada social structure probably act as a deterrent to predators [48], large units may monopolize more central locations within the band during the day or sleeping positions on the cliffs at night, thus leaving small units vulnerable at the periphery. This question will require simultaneous targeted data collection on relative group position—a focus of our future research.

What, if anything, can females in suboptimal groups do? In broad strokes, they have three theoretical choices: they can disperse to a new unit entirely, they can add females to their too-small units or they can subtract females from their too-large units. In this population, female dispersal is rare ($n = 2$ across the entire study period; 0.008 per unit-year); reproductive units consist of close maternal kin, and females tend to remain in the same unit they were born into [71]. Adding females can be accomplished through demographic processes (maturation); however, small units, given their lower reproductive performance and high adult female death rates, require more time to grow than mid-sized or large units. Some small units may never reach an optimal size, or, worse, could become 'sinks', at risk of dying out entirely (and we have seen at least one of our units go extinct). This raises the possibility that unit size and female fitness are bidirectionally related: units grow or shrink according to female reproductive performance and survival, while group size itself can also impose fitness costs, creating a feedback loop. Therefore, for some small units, a more effective way to add females may involve joining with another unit (fusions). Fusions are rare ($n = 3$ across the entire study period; 0.012 per unit-year) and may involve a number of hurdles (e.g. resistance of leader males and/or unrelated females), but these observations suggest that this is nevertheless a viable option for gelada females.

Although some small units may become sinks, overall demographic analyses from this population indicate that the stochastic growth rate is 3.5% [72]. Thus, the more common problem (for mid-sized and large units) is that they continue to grow in size. Subtracting females from units can occur when one group splits permanently into two groups (fissions), which we routinely observe for large units ($n = 11$; 0.045 per unit-year). Compared to other primate taxa, gelada social organization may offer more opportunities for fissions to occur, as the band may buffer the ecological costs of fissions. In sum, unit size changes occur via frequent, slow and small demographic processes (i.e. maturations and deaths) combined with rare, rapid and large group dynamics (i.e. fusions and fissions). Future analyses will focus on identifying the relative importance that these events have in structuring groups and the social network metrics that anticipate changes in group dynamics.

Takeover frequency was positively associated with the number of females in a unit (as previously reported in this population [52]) but was not associated with the number of males in a unit. This stands in contrast with our previous research showing that multi-male units (consisting of a leader male and 1+ follower males) experienced fewer takeovers per unit-year than units with only one adult leader male across all unit sizes [47]. In geladas, the consequences of males for female fitness may depend on the reproductive strategies of individual follower males and on the ability of leader males to regulate group membership (rather than solely on the quantity of males), which require further data and analysis. Indeed, in a number of primates, group composition (e.g. a higher ratio of males to females) has been shown to reduce the risk of takeovers and/or infanticide (e.g. [73]; reviewed in [74]). For example, the reproductive success of female white-faced capuchins (*Cebus capucinus*) was both negatively correlated with group size and positively correlated with the ratio of males to females in the group [75]. By contrast, multi-male groups of ursine colobus monkeys (*Colobus vellerosus*) had higher rates of male immigration and infanticide, perhaps because these dominant males were ineffective at deterring immigrants [40]. Whether female geladas in small or large *multi-male* units demonstrate higher reproductive performance than females in small or large *single-male* units remains to be seen.

The link between group size and individual fitness has long been the focus of socioecological theory, yet it has been difficult to demonstrate this relationship for long-lived species with empirical data. Here, we leveraged long-term data from a population of wild geladas to provide insight into how and why group size shapes female fitness—specifically, adult female deaths and conspecific infanticide.

**Data accessibility.** All data and code used in this analysis are available at: https://doi.org/10.5281/zenodo.4737602 [76].

**Authors' contributions.** E.T.J.: conceptualization, data curation, formal analysis, funding acquisition, methodology, visualization, writing-original draft, writing-review and editing; J.A.F.: data curation, formal analysis, funding acquisition, visualization, writing-original draft, writing-review and editing; T.J.B.: conceptualization, formal analysis, funding acquisition, investigation, methodology, project administration, supervision, writing-review and editing; A.L.: data curation, formal analysis, funding acquisition, investigation, methodology, project administration, supervision, writing-review and editing; N.S.-M.: conceptualization, data curation, formal analysis, funding acquisition, investigation, methodology, project administration, supervision, visualization, writing-original draft, writing-review and editing; J.C.B.: conceptualization, data curation, formal analysis, funding acquisition, investigation, methodology, project administration, supervision, visualization, writing-review and editing.

All authors gave final approval for publication and agreed to be held accountable for the work performed therein.

**Competing interests.** Authors declare no competing interests.

**Funding.** This work was supported by National Science Foundation (grant nos. BCS-0715179, IOS-1255974, BCS-1340911, IOS-1854359; BCS-1723237; BCS-17123228; BCS-2010309); Leakey Foundation (multiple grants); National Institutes of Health (grant no. R00-AG051764); National Geographic Society (grant nos. 8100–06, 8989–11, 50409R-18; the Fulbright Scholar Program; Sigma Xi; American Society of Primatologists; Wildlife Conservation Society and the University of Michigan.

**Acknowledgements.** We would like to thank the Ethiopian Wildlife Conservation Authority (EWCA) and the wardens and staff of the Simien Mountain National Park for their permission and ongoing support for our long-term research project. Additionally, we are grateful to our excellent field team across the years, most especially E. Jejaw, A. Fanta, S. Girmay, J. Jarvey and M. Gomery for their assistance with field data collection. We also owe thanks to A. Marshall, B. Dantzer, J. Mitani and two anonymous reviewers for providing valuable feedback on this manuscript.

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
