## [Peer Review File · Proceedings of the Royal Society B: Biological Sciences]

Review History

RSPB-2021-0313.R0 (Original submission)

Review form: Reviewer 1

Recommendation

Major revision is needed (please make suggestions in comments)

Scientific importance: Is the manuscript an original and important contribution to its field?

Excellent

General interest: Is the paper of sufficient general interest?

Excellent

Quality of the paper: Is the overall quality of the paper suitable?

Excellent

Is the length of the paper justified?

Yes

Should the paper be seen by a specialist statistical reviewer?

No

Do you have any concerns about statistical analyses in this paper? If so, please specify them explicitly in your report.

No

It is a condition of publication that authors make their supporting data, code and materials available - either as supplementary material or hosted in an external repository. Please rate, if applicable, the supporting data on the following criteria.

Is it accessible?

Yes

Is it clear?

Yes

Is it adequate?

Yes

Do you have any ethical concerns with this paper?

No

Comments to the Author

This manuscript presents a detailed investigation of the relationships between group size and various aspects of female reproductive success in geladas. The analyses are based on an impressive dataset (in particular for such a long-lived primate) and solid - and it is great that the authors share the data and code necessary to repeat the analyses. The findings help advance our understanding of how different factors, in particular social ones, might interact to form selection for individuals to be in optimal group sizes. While the analyses cannot identify exactly how and why there are fitness differences associated with the number of other females in the group, they provide a base from which to start such investigations. I repeatedly caught myself thinking "but what about this [e/g smaller groups might be at the periphery]", but realised that this research lays the foundation to explore such more specific questions. I only have a few questions and comments about decisions in the statistical analyses, about the overall fitness of females, and about the interpretation. Even though I chose "major revision", I think that it should be possible to address all my comments relatively easily by editing some passage to provide additional information.

- Decisions in the statistical analyses

The analyses of survival of females are based on whether a female died in a year. Does this refer to Jan 01 - Dec 31st? For the group size in that year, does this mean averaging across the months during which a female was alive?

Why did you not use the Cox survival model for the adult females? Would this not also more directly provide you with the expected reproductive lifespan?

You should mention at each section of the results that the respective full model output can be found in the supplement, you currently only refer to Table S6.

Could you show or describe the group size distribution? That is, what percentage of females (female per year) are found alone, with one other female, two other...11 other females? It would help to put these findings into context - for example, it would indicate whether, for example, despite fitness being highest at intermediate group sizes, most offspring are born to females in large groups.

You describe that average ages of females are not different in groups of different sizes. Is there maybe though a variance effect - that is, when groups are started small when females of average

age fission, and as these groups grow, these females age and new young females are recruited. In this case, the average age would not change, but old females would only be found in large groups (e.g. group starts with two females of age 8, four years later these two females are age 12 and there are two daughters of age 4 so the average age is still 8).

- Overall fitness of females

In the section on reproductive performance, you mention the differences in the expected number of offspring females in different groups might produce. I wonder whether you could expand on this because it is the key part to the interpretation in the discussion. I think there could be a more formal, demography based approach to model life-time fitness, but this is probably beyond the scope of this article. I think it would however be helpful if you could calculate expected numbers of offspring across the range of group sizes.

In particular, this would help to the understanding of why females might be in group sizes larger than the optimum, which is the expectation from most optimal group size models (e.g. your reference 21). I think your data indicates that being in a small group is relatively worse than being in a large group - so, fitness in group of 8 is better than fitness in group of 4 even though both are 2 away from the optimum group size of 6. Because of this fitness skew, groups will not split at size 8 such that many large groups exist, which appears to be the only way for females to actively regulate group size.

You do not mention age at first reproduction, which could have an additional influence on overall female reproductive success (eg. if age at first reproduction is 2 years later in groups of certain sizes this would mean 20% lower reproductive success). Do you think this differs across group sizes?

- Framing

I think that in general you are careful in not attributing cause-effect interpretations to the findings. However, the discussion only focuses on group sizes potentially shaping female reproduction (e.g. line 433f "Our findings provide insight into how and why group size shapes female fitness"). The issue is that this causation is also reversed: as you explain, group size is directly (and almost exclusively) determined by recruitment of females born in the same group and by mortality of females. For example, if certain groups experience high mortality because they are at the periphery of the large bands, their group size cannot increase (there might actually be source-sink effects where large groups continuously grow and fission and small groups go extinct). I think this needs to be mentioned in the discussion.

Review form: Reviewer 2

Recommendation

Major revision is needed (please make suggestions in comments)

Scientific importance: Is the manuscript an original and important contribution to its field?

Good

General interest: Is the paper of sufficient general interest?

Good

Quality of the paper: Is the overall quality of the paper suitable?

Good

Is the length of the paper justified?

Yes

Should the paper be seen by a specialist statistical reviewer?

No

Do you have any concerns about statistical analyses in this paper? If so, please specify them explicitly in your report.

Yes

It is a condition of publication that authors make their supporting data, code and materials available - either as supplementary material or hosted in an external repository. Please rate, if applicable, the supporting data on the following criteria.

Is it accessible?

Yes

Is it clear?

Yes

Is it adequate?

Yes

Do you have any ethical concerns with this paper?

No

Comments to the Author

This manuscript describes the gelada goldilocks effect, where females have higher fitness under intermediate unit sizes. The team uses an impressive long-term dataset to explore how female mortality, number of young that survive to weaning, or inter-birth interval varied with female unit size (ranging from 1 to 12). The study finds that the goldilocks effect is driven by female mortality increasing as unit size decreased while male takeovers and associated infanticide risk increased with unit size. Thus, females from intermediate groups should have a fitness advantage. I found this study to be on an interesting topic and to contribute novel empirical evidence to support a relationship between fitness and group living – surprisingly, there are few empirical studies that test the relationship between social group size and fitness, despite being a fundamental question on the evolution of sociality. I feel this manuscript could make a strong contribution to the literature, though I did have some concerns about the language and methods in the manuscript's current form:

1. My first concern is about the calculations made to estimate the lifetime reproductive success of female geladas. As I understand it, the authors calculated mean lifespan for females from small, medium and large groups. The authors also calculated the mean conception rate of females for small, medium and large groups, and then multiplied the two to estimate LRS. I think a number of issues exist with this approach. First, these means include uncertainty in their calculations that are not being carried forward in the calculation. Second, I think it is very likely given that females experience different selective pressures in different sized units that individual females may invest in reproduction differently throughout their lives and extrapolating reproductive rates this way does not take different reproductive strategies into account (this assumes that reproductive rates are fixed over the life of an animal when age can certainly influence this a lot, for example). Finally, I'm not sure I fully understand the value of making these calculations rather than sticking to the data, or calculating LRS on a subset of females that have been observed over their whole lives.
2. My second concern is about some of the language in the manuscript which I found confusing in many places. I pointed out these areas in the comments below, but two recurring instances included (i) "mortality" being used for both adult females, as well as infants, and (ii) "female success", "infant success" and "success" being used throughout (perhaps

interchangeably?) where I think you simply mean to say infant surviving to weaning. Overall, I enjoyed the manuscript and found the results and story compelling. I include questions and suggested clarifications below. I wish the authors luck in revising this work.

Comments:

Line 41: I think the costs and benefits of group living could be made a bit more general here when you introduce them. Competition when foraging and predation risk are two examples that apply to your study system, but there are many others like competition for or access to mates, communal resource defence, thermoregulation, cooperative breeding/parental care.

Line 70: It's important to distinguish between within and among-group competition when discussing social species. I believe here you mean within-group feeding competition? But in some social species larger groups better defend food resources and thus have reduced competition for food from belonging to larger units.

Line 72: Here, I think it gets confusing because you must have female fitness in mind specifically but I thought this was still a general section on group size and fitness. In your example with infanticide risk, I assume males would have higher reproductive success if they took over a unit with more females, no? Infanticide should increase male fitness but would decrease female fitness. Perhaps discussing conflict in optimal group size between males and females would help make your point?

Line 125: how often is each unit seen/observed? A range of lengths of time would be useful for those not familiar with the system.

Line 138: If follower males can deter takeovers (and possible infanticide), wouldn't the number of follower males in a group also benefit females? For example, large female units with more follower males can be shielded from takeovers? Is it possible to look at the interaction between female and male group sizes on female mortality or reproductive success?

Line 172: Why did you perform the female and infant mortality analyses differently? Why not run survival models for both?

Line 180 (also 203, and others): Are these separate random effects, or were they nested in any way?

Line 199: "monthly success" is a bit confusing here. I assume you refer to the success of weaning >1 offspring? Perhaps clarify with a more specific term.

Line 204: What is the logic behind using 6-month vs monthly bins? Do geladas have distinct breeding seasons, or can females breed year-round? You could clarify what the typical reproductive cycle is within a unit or within one female and how your temporal bins map on to the biology of the system?

Line 212: Are "successful" offspring the same thing offspring that survived until weaning? I'm not sure if successful is a common term used in primatology research to mean weaned young, but I find it confusing throughout the paper as different researchers may identify "success" with different developmental points.

Line 248: Why would you need to control/offset for total observation time at each unit? Are takeovers so quick or short-lived that units that are less frequently observed are at risk of missing takeover events? My understanding of a takeover is that the dominant male of a unit becomes permanently displaced by a new male, which seems like a rather permanent event. Please clarify whether I am mistaken, or why controlling for total observation time is required.

Line 276: How does reproductive lifespan differ from lifespan? Are you somehow incorporating reproductive senescence into these estimates, and if so how?

Line 285: Are you multiplying the average lifespan of females in different sized units by the average annual reproductive rate across units? This is likely misleading as a measure of lifetime reproductive success as both averages have uncertainty associated with them that is not being carried forward into these calculations. This broad unit-level calculation doesn't incorporate important information about how unit size might shape reproductive decisions or possible competition within a unit. For example, in some social animals variance in reproductive success differs with group size. I would include caveats about this type of calculation up front, at the very least. Otherwise, I'm not sure I trust these numbers and don't understand the purpose of these calculations.

Figure 2A: Can you include post-hoc indications of which group is different from which group

rather than the p-value? I cannot tell which means differ from one another based on the figure.
Line 358: can you clarify whether you mean adult or offspring mortality? It is a bit confusing when you have two mortality variables.

Line 365: I'm not sure I agree with your projections based on unit-level means multiplied by one another. I would argue that females may be making different reproductive decisions in different sized units based on the pressures they face, particularly if they face different risks of infant mortality as your results show. Without more information on reproductive investment across different sized units I'm not convinced the broad estimates hold true when you extrapolate to lifetime levels. You must have lifetime data for some individuals – in this subset of your data, can you compare lifetime reproductive success across units of different sizes to support or replace these calculations?

Line 377: "infanticide" typo

Supplemental tables: Please report random effects

Decision letter (RSPB-2021-0313.R0)

19-Mar-2021

Dear Dr Tinsley Johnson:

I am writing to inform you that your manuscript RSPB-2021-0313 entitled "The Goldilocks Effect: Female geladas in mid-sized groups have higher fitness" has, in its current form, been rejected for publication in Proceedings B.

This action has been taken on the advice of referees, who have recommended that substantial revisions are necessary. With this in mind we would be happy to consider a resubmission, provided the comments of the referees are fully addressed. However please note that this is not a provisional acceptance.

Sincerely,
 Dr Sasha Dall
 mailto: proceedingsb@royalsociety.org

Reviewer(s)' Comments to Author:

Referee: 1

Comments to the Author(s)

This manuscript presents a detailed investigation of the relationships between group size and various aspects of female reproductive success in geladas. The analyses are based on an impressive dataset (in particular for such a long-lived primate) and solid - and it is great that the authors share the data and code necessary to repeat the analyses. The findings help advance our understanding of how different factors, in particular social ones, might interact to form selection for individuals to be in optimal group sizes. While the analyses cannot identify exactly how and why there are fitness differences associated with the number of other females in the group, they provide a base from which to start such investigations. I repeatedly caught myself thinking "but what about this [e/g smaller groups might be at the periphery]", but realised that this research lays the foundation to explore such more specific questions. I only have a few questions and comments about decisions in the statistical analyses, about the overall fitness of females, and about the interpretation. Even though I chose "major revision", I think that it should be possible to address all my comments relatively easily by editing some passage to provide additional information.

- Decisions in the statistical analyses

The analyses of survival of females are based on whether a female died in a year. Does this refer to Jan 01 - Dec 31st? For the group size in that year, does this mean averaging across the months during which a female was alive?

Why did you not use the Cox survival model for the adult females? Would this not also more directly provide you with the expected reproductive lifespan?

You should mention at each section of the results that the respective full model output can be found in the supplement, you currently only refer to Table S6.

Could you show or describe the group size distribution? That is, what percentage of females (female per year) are found alone, with one other female, two other...11 other females? It would help to put these findings into context - for example, it would indicate whether, for example, despite fitness being highest at intermediate group sizes, most offspring are born to females in large groups.

You describe that average ages of females are not different in groups of different sizes. Is there maybe though a variance effect - that is, when groups are started small when females of average age fission, and as these groups grow, these females age and new young females are recruited. In this case, the average age would not change, but old females would only be found in large groups (e.g. group starts with two females of age 8, four years later these two females are age 12 and there are two daughters of age 4 so the average age is still 8).

- Overall fitness of females

In the section on reproductive performance, you mention the differences in the expected number of offspring females in different groups might produce. I wonder whether you could expand on this because it is the key part to the interpretation in the discussion. I think there could be a more formal, demography based approach to model life-time fitness, but this is probably beyond the scope of this article. I think it would however be helpful if you could calculate expected numbers of offspring across the range of group sizes.

In particular, this would help to the understanding of why females might be in group sizes larger than the optimum, which is the expectation from most optimal group size models (e.g. your reference 21). I think your data indicates that being in a small group is relatively worse than being in a large group - so, fitness in group of 8 is better than fitness in group of 4 even though both are 2 away from the optimum group size of 6. Because of this fitness skew, groups will not split at size 8 such that many large groups exist, which appears to be the only way for females to actively regulate group size.

You do not mention age at first reproduction, which could have an additional influence on overall female reproductive success (eg. if age at first reproduction is 2 years later in groups of certain sizes this would mean 20% lower reproductive success). Do you think this differs across group sizes?

- Framing

I think that in general you are careful in not attributing cause-effect interpretations to the findings. However, the discussion only focuses on group sizes potentially shaping female reproduction (e.g. line 433f "Our findings provide insight into how and why group size shapes female fitness"). The issue is that this causation is also reversed: as you explain, group size is directly (and almost exclusively) determined by recruitment of females born in the same group and by mortality of females. For example, if certain groups experience high mortality because they are at the periphery of the large bands, their group size cannot increase (there might actually be source-sink effects where large groups continuously grow and fission and small groups go extinct). I think this needs to be mentioned in the discussion.

Referee: 2

Comments to the Author(s)

This manuscript describes the gelada goldilocks effect, where females have higher fitness under intermediate unit sizes. The team uses an impressive long-term dataset to explore how female mortality, number of young that survive to weaning, or inter-birth interval varied with female unit size (ranging from 1 to 12). The study finds that the goldilocks effect is driven by female mortality increasing as unit size decreased while male takeovers and associated infanticide risk increased with unit size. Thus, females from intermediate groups should have a fitness advantage. I found this study to be on an interesting topic and to contribute novel empirical evidence to support a relationship between fitness and group living – surprisingly, there are few empirical studies that test the relationship between social group size and fitness, despite being a fundamental question on the evolution of sociality. I feel this manuscript could make a strong contribution to the literature, though I did have some concerns about the language and methods in the manuscript's current form:

1. My first concern is about the calculations made to estimate the lifetime reproductive success of female geladas. As I understand it, the authors calculated mean lifespan for females from small, medium and large groups. The authors also calculated the mean conception rate of females for small, medium and large groups, and then multiplied the two to estimate LRS. I think a number of issues exist with this approach. First, these means include uncertainty in their calculations that are not being carried forward in the calculation. Second, I think it is very likely given that females experience different selective pressures in different sized units that individual females may invest in reproduction differently throughout their lives and extrapolating reproductive rates this way does not take different reproductive strategies into account (this assumes that reproductive rates are fixed over the life of an animal when age can certainly influence this a lot, for example). Finally, I'm not sure I fully understand the value of making these calculations rather than sticking to the data, or calculating LRS on a subset of females that have been observed over their whole lives.

2. My second concern is about some of the language in the manuscript which I found confusing in many places. I pointed out these areas in the comments below, but two recurring instances included (i) "mortality" being used for both adult females, as well as infants, and (ii) "female

success", "infant success" and "success" being used throughout (perhaps interchangeably?) where I think you simply mean to say infant surviving to weaning.

Overall, I enjoyed the manuscript and found the results and story compelling. I include questions and suggested clarifications below. I wish the authors luck in revising this work.

Comments:

Line 41: I think the costs and benefits of group living could be made a bit more general here when you introduce them. Competition when foraging and predation risk are two examples that apply to your study system, but there are many others like competition for or access to mates, communal resource defence, thermoregulation, cooperative breeding/parental care.

Line 70: It's important to distinguish between within and among-group competition when discussing social species. I believe here you mean within-group feeding competition? But in some social species larger groups better defend food resources and thus have reduced competition for food from belonging to larger units.

Line 72: Here, I think it gets confusing because you must have female fitness in mind specifically but I thought this was still a general section on group size and fitness. In your example with infanticide risk, I assume males would have higher reproductive success if they took over a unit with more females, no? Infanticide should increase male fitness but would decrease female fitness. Perhaps discussing conflict in optimal group size between males and females would help make your point?

Line 125: how often is each unit seen/observed? A range of lengths of time would be useful for those not familiar with the system.

Line 138: If follower males can deter takeovers (and possible infanticide), wouldn't the number of follower males in a group also benefit females? For example, large female units with more follower males can be shielded from takeovers? Is it possible to look at the interaction between female and male group sizes on female mortality or reproductive success?

Line 172: Why did you perform the female and infant mortality analyses differently? Why not run survival models for both?

Line 180 (also 203, and others): Are these separate random effects, or were they nested in any way?

Line 199: "monthly success" is a bit confusing here. I assume you refer to the success of weaning >1 offspring? Perhaps clarify with a more specific term.

Line 204: What is the logic behind using 6-month vs monthly bins? Do geladas have distinct breeding seasons, or can females breed year-round? You could clarify what the typical reproductive cycle is within a unit or within one female and how your temporal bins map on to the biology of the system?

Line 212: Are "successful" offspring the same thing offspring that survived until weaning? I'm not sure if successful is a common term used in primatology research to mean weaned young, but I find it confusing throughout the paper as different researchers may identify "success" with different developmental points.

Line 248: Why would you need to control/offset for total observation time at each unit? Are takeovers so quick or short-lived that units that are less frequently observed are at risk of missing takeover events? My understanding of a takeover is that the dominant male of a unit becomes permanently displaced by a new male, which seems like a rather permanent event. Please clarify whether I am mistaken, or why controlling for total observation time is required.

Line 276: How does reproductive lifespan differ from lifespan? Are you somehow incorporating reproductive senescence into these estimates, and if so how?

Line 285: Are you multiplying the average lifespan of females in different sized units by the average annual reproductive rate across units? This is likely misleading as a measure of lifetime reproductive success as both averages have uncertainty associated with them that is not being carried forward into these calculations. This broad unit-level calculation doesn't incorporate important information about how unit size might shape reproductive decisions or possible competition within a unit. For example, in some social animals variance in reproductive success differs with group size. I would include caveats about this type of calculation up front, at the very least. Otherwise, I'm not sure I trust these numbers and don't understand the purpose of these calculations.

Figure 2A: Can you include post-hoc indications of which group is different from which group rather than the p-value? I cannot tell which means differ from one another based on the figure.
Line 358: can you clarify whether you mean adult or offspring mortality? It is a bit confusing when you have two mortality variables.

Line 365: I'm not sure I agree with your projections based on unit-level means multiplied by one another. I would argue that females may be making different reproductive decisions in different sized units based on the pressures they face, particularly if they face different risks of infant mortality as your results show. Without more information on reproductive investment across different sized units I'm not convinced the broad estimates hold true when you extrapolate to lifetime levels. You must have lifetime data for some individuals – in this subset of your data, can you compare lifetime reproductive success across units of different sizes to support or replace these calculations?

Line 377: “infanticide” typo

Supplemental tables: Please report random effects

Author's Response to Decision Letter for (RSPB-2021-0313.R0)

See Appendix A.

RSPB-2021-0820.R0

Review form: Reviewer 1

Recommendation

Accept with minor revision (please list in comments)

Scientific importance: Is the manuscript an original and important contribution to its field?

Good

General interest: Is the paper of sufficient general interest?

Excellent

Quality of the paper: Is the overall quality of the paper suitable?

Excellent

Is the length of the paper justified?

Yes

Should the paper be seen by a specialist statistical reviewer?

No

Do you have any concerns about statistical analyses in this paper? If so, please specify them explicitly in your report.

No

It is a condition of publication that authors make their supporting data, code and materials available - either as supplementary material or hosted in an external repository. Please rate, if applicable, the supporting data on the following criteria.

Is it accessible?

No

Is it clear?

N/A

Is it adequate?

N/A

Do you have any ethical concerns with this paper?

No

Comments to the Author

The authors have done a great job in addressing my comments. The revised version provides helpful additional context and points out how future research might be built on the described findings. I only have two minor comments:

In the revised version you report that the death rate of adult individuals in small groups is 10.4% and in large groups 7.3% - but in both small and large groups you estimate that individuals have median age at death of 16 years (line 292). So, for a given age, is there higher mortality in large groups, but because there are more young individuals at start of group in the larger group, the expected lifespan is the same?

- It appears that during the latest update of the GitHub repository the data file was deleted (there are now two code scripts, one apparently the original another revised). I went to the earlier version of the repository and could download the file, so everything seems to be there. I think it would be good to update this. My suggestion would also be to deposit the data at a repository with a DOI so that it is more permanently available (avoiding the risk that a collaborator closes their GitHub repository). I can recommend KNB, which is free and provides helpful settings to add metadata for ecological datasets (which increases discoverability) and add a licence (which clarifies and increases reusability): <https://knb.ecoinformatics.org>

The code does refer to an additional input data file that does not seem to be part of the rdata package - the file cyphers_dobs.csv which appears to be used to assign dates of births to females who were alive at the beginning of the study. Is this preprocessing and the resulting output is part of the rdata package?

Review form: Reviewer 2**Recommendation**

Accept as is

Scientific importance: Is the manuscript an original and important contribution to its field?

Good

General interest: Is the paper of sufficient general interest?

Excellent

Quality of the paper: Is the overall quality of the paper suitable?

Good

Is the length of the paper justified?

Yes

Should the paper be seen by a specialist statistical reviewer?

No

Do you have any concerns about statistical analyses in this paper? If so, please specify them explicitly in your report.

No

It is a condition of publication that authors make their supporting data, code and materials available - either as supplementary material or hosted in an external repository. Please rate, if applicable, the supporting data on the following criteria.

Is it accessible?

Yes

Is it clear?

Yes

Is it adequate?

Yes

Do you have any ethical concerns with this paper?

No

Comments to the Author

I previously reviewed this manuscript, and I thank the authors for their work in revising and explaining their research in the resubmission. I think the manuscript is generally much clearer and represents important work. Nicely done.

Decision letter (RSPB-2021-0820.R0)

04-May-2021

Dear Dr Tinsley Johnson

I am pleased to inform you that your manuscript RSPB-2021-0820 entitled "The Goldilocks Effect: Female geladas in mid-sized groups have higher fitness" has been accepted for publication in Proceedings B.

The referee(s) have recommended publication, but also suggest some minor revisions to your manuscript. Therefore, I invite you to respond to the referee(s)' comments and revise your manuscript. Because the schedule for publication is very tight, it is a condition of publication that you submit the revised version of your manuscript within 7 days. If you do not think you will be able to meet this date please let us know.

When submitting your revised manuscript, you will be able to respond to the comments made by the referee(s) and upload a file "Response to Referees". You can use this to document any changes

you make to the original manuscript. We require a copy of the manuscript with revisions made since the previous version marked as 'tracked changes' to be included in the 'response to referees' document.

Sincerely,
Dr Sasha Dall
mailto: proceedingsb@royalsociety.org

Associate Editor
Board Member

Comments to Author:

The reviewers and I agree that you have addressed all of the comments and the manuscript will make a very nice contribution to the field. There are only two very minor clarifications to fix- one wording clarification and fixing the file in the GitHub repository.

Reviewer(s)' Comments to Author:

Referee: 1

Comments to the Author(s).

The authors have done a great job in addressing my comments. The revised version provides helpful additional context and points out how future research might built on the described findings. I only have two minor comments:

In the revised version you report that the death rate of adult individuals in small groups is 10.4% and in large groups 7.3% - but in both small and large groups you estimate that individuals have median age at death of 16 years (line 292). So, for a given age, is there higher mortality in large groups, but because there are more young individuals at start of group in the larger group, the expected lifespan is the same?

- It appears that during the latest update of the GitHub repository the data file was deleted (there are now two code scripts, one apparently the original another revised). I went to the earlier version of the repository and could download the file, so everything seems to be there. I think it would be good to update this. My suggestion would also be to deposit the data at a repository with a DOI so that it is more permanently available (avoiding the risk that a collaborator closes their GitHub repository). I can recommend KNB, which is free and provides helpful settings to add metadata for ecological datasets (which increases discoverability) and add a licence (which clarifies and increases reusability): <https://knb.ecoinformatics.org>

The code does refer to an additional input data file that does not seem to be part of the rdata package - the file cyphers_dobs.csv which appears to be used to assign dates of births to females who were alive at the beginning of the study. Is this preprocessing and the resulting output is part of the rdata package?

Referee: 2

Comments to the Author(s).

I previously reviewed this manuscript, and I thank the authors for their work in revising and explaining their research in the resubmission. I think the manuscript is generally much clearer and represents important work. Nicely done.

Author's Response to Decision Letter for (RSPB-2021-0820.R0)

See Appendix B.

Decision letter (RSPB-2021-0820.R1)

05-May-2021

Dear Dr Tinsley Johnson

I am pleased to inform you that your manuscript entitled "The Goldilocks Effect: Female geladas in mid-sized groups have higher fitness" has been accepted for publication in Proceedings B.

Data Accessibility section

Open Access

Paper charges

Sincerely,

Proceedings B

Appendix A

19-Mar-2021

Dear Dr Tinsley Johnson:

I am writing to inform you that your manuscript RSPB-2021-0313 entitled "The Goldilocks Effect: Female geladas in mid-sized groups have higher fitness" has, in its current form, been rejected for publication in Proceedings B.

This action has been taken on the advice of referees, who have recommended that substantial revisions are necessary. With this in mind we would be happy to consider a resubmission, provided the comments of the referees are fully addressed. However please note that this is not a provisional acceptance.

Sincerely,

Dr Sasha Dall
mailto:proceedingsb@royalsociety.org

Reviewer(s)' Comments to Author:

Referee: 1

Comments to the Author(s)

This manuscript presents a detailed investigation of the relationships between group size and various aspects of female reproductive success in geladas. The analyses are based on an impressive dataset (in particular for such a long-lived primate) and solid - and it is great that the authors share the data and code necessary to repeat the analyses. The findings help advance our understanding of how different factors, in particular social ones, might interact to form selection for individuals to be in optimal group sizes. While the analyses cannot identify exactly how and why there are fitness differences associated with the number of other females in the group, they provide a base from which to start such investigations. I repeatedly caught myself thinking "but what about this [e/g smaller groups might be at the periphery]", but realised that this research lays the foundation to explore such more specific questions. I only have a few questions and comments about decisions in the statistical analyses, about the overall fitness of females, and about the interpretation. Even though I chose "major revision", I think that it should be possible to address all my comments relatively easily by editing some passages to provide additional information.

REVIEWER COMMENT	REPLY
Decisions in the statistical analyses	
The analyses of survival of females are based on whether a female died in a year. Does this refer to Jan 01 - Dec 31st? For the group size in that year, does this mean averaging across the months during which a female was alive?	Yes, years were counted from Jan 01 - Dec 31 or averaged across the months a female was alive (if the female died mid-year) or mature (if a female matured mid-year). We have clarified this in L178-179.
Why did you not use the Cox survival model for the adult females? Would this not also more directly provide you with the expected reproductive lifespan?	We initially did not use Cox survival models for both adults and infants, as the ages for many adult females are coarsely estimated (only 56/200 females had known dates of birth). Given this uncertainty, we had decided it would be more conservative to estimate death rates while controlling for differences in age (known or estimated), whereas age is explicitly embedded in the structure of survival models. In light of this suggestion, however, we now provide the results from Cox mixed effects models of adult female survival, for which we

	incorporated uncertainty in female ages. These model results are included in the supplementary materials.
You should mention at each section of the results that the respective full model output can be found in the supplement, you currently only refer to Table S6.	We now reference the supplementary tables throughout the Results.
Could you show or describe the group size distribution? That is, what percentage of females (female per year) are found alone, with one other female, two other...11 other females? It would help to put these findings into context - for example, it would indicate whether, for example, despite fitness being highest at intermediate group sizes, most offspring are born to females in large groups.	We now present a histogram of female unit sizes (Fig S1), measured using the number of female-years at each unit size.
You describe that average ages of females are not different in groups of different sizes. Is there maybe though a variance effect - that is, when groups are started small when females of average age fission, and as these groups grow, these females age and new young females are recruited. In this case, the average age would not change, but old females would only be found in large groups (e.g. group starts with two females of age 8, four years later these two females are age 12 and there are two daughters of age 4 so the average age is still 8).	This is an excellent point! Given the results for adult female mortality, it would follow that medium-sized and large units would contain more old females. However, density plots of female age and Kolmogorov-Smirnov tests across the three unit size categories indicate that large units contain more young females, perhaps due to a larger pool for juvenile recruitment. We now mention this in the text (L170-174) and include this in Figure S2.
Overall fitness of females	
In the section on reproductive performance, you mention the differences in the expected number of offspring females in different groups might produce. I wonder whether you could expand on this because it is the key part to the interpretation in the discussion. I think there could be a more formal, demography based approach to model life-	We have now added caveats to this in the Methods (L228-232) and in Discussion (L401-403). Given that unit sizes are not stable across females' lifespans, these calculations suggest the potential for fitness differences across unit sizes as well as the potential need for strategies to mitigate these effects (e.g., fissions), which may not translate to true lifetime fitness differences. The analysis mentioned here (now L220-232)

time fitness, but this is probably beyond the scope of this article.

I think it would however be helpful if you could calculate expected numbers of offspring across the range of group sizes. In particular, this would help to the understanding of why females might be in group sizes larger than the optimum, which is the expectation from most optimal group size models (e.g. your reference 21). I think your data indicates that being in a small group is relatively worse than being in a large group - so, fitness in group of 8 is better than fitness in group of 4 even though both are 2 away from the optimum group size of 6. Because of this fitness skew, groups will not split at size 8 such that many large groups exist, which appears to be the only way for females to actively regulate group size.

was, as suggested, a coarse estimation using the data we currently have, but definitely something we would like to follow-up on with a more formal, demography-based approach in the future. In an effort to more directly reflect our current data, we now estimate reproductive lifespans by subtracting the population mean age at first birth from the median age at death at each unit size category, accounting for right-censored data (L222-224).

We agree that presenting estimates across the entire range of unit sizes would be ideal. However, given smaller sample sizes and fluctuations in unit size within females' lifetimes, estimating median survival at each unit size is not currently feasible.

You do not mention age at first reproduction, which could have an additional influence on overall female reproductive success (eg. if age at first reproduction is 2 years later in groups of certain sizes this would mean 20% lower reproductive success). Do you think this differs across group sizes?

Good point! We now discuss how a "Vandenbergh effect" (the presence/absence of a non-sire male influences maturation in young females: Lu et al., 2021) might interact with takeover frequencies across unit sizes. Specifically, young females in small units might delay maturation (to avoid inbreeding) and young females in larger units might mature earlier. However, there is overall a very narrow range in variation of age at first birth (a 2-year range between the ages of 5-7 for 90%+ of females in our study population). This, combined with previous research in similarly long-lived primates showing age at first birth had a nominal effect on lifetime reproductive success, suggests that differences in age at first birth would not have substantial effects on our lifetime fitness estimations.

We now address this in our Discussion (L391-393) and emphasize that our lifetime reproductive success estimation is coarse and does not take age at first birth into account.

Framing

I think that in general you are careful in not attributing cause-effect interpretations to the findings. However, the discussion only focuses on group sizes potentially shaping female reproduction (e.g. line 433f "Our findings provide insight into how and why group size shapes female fitness"). The issue is that this causation is also reversed: as you explain, group size is directly (and almost exclusively) determined by recruitment of females born in the same group and by mortality of females. For example, if certain groups experience high mortality because they are at the periphery of the large bands, their group size cannot increase (there might actually be source-sink effects where large groups continuously grow and fission and small groups go extinct). I think this needs to be mentioned in the discussion.

Thank you -- we have added this to our Discussion (L427-L430). We do agree that there is a bidirectional relationship in that unit size shapes female fitness and female fitness shapes unit size; however, we also suspect that there are more drastic strategies for altering unit size (i.e., fissions and fusions) that have a greater impact on unit size -- the focus of a future analysis.

Referee: 2

Comments to the Author(s)

This manuscript describes the gelada goldilocks effect, where females have higher fitness under intermediate unit sizes. The team uses an impressive long-term dataset to explore how female mortality, number of young that survive to weaning, or inter-birth interval varied with female unit size (ranging from 1 to 12). The study finds that the goldilocks effect is driven by female mortality increasing as unit size decreased while male takeovers and associated infanticide risk increased with unit size. Thus, females from intermediate groups should have a fitness advantage. I found this study to be on an interesting topic and to contribute novel empirical evidence to support a relationship between fitness and group living—surprisingly, there are few empirical studies that test the relationship between social group size and fitness, despite being a fundamental question on the evolution of sociality.

I feel this manuscript could make a strong contribution to the literature, though I did have some concerns about the language and methods in the manuscript's current form:

1. My first concern is about the calculations made to estimate the lifetime reproductive success of female geladas. As I understand it, the authors calculated mean lifespan for females from small, medium and large groups. The authors also calculated the mean conception rate of females for small, medium and large groups, and then multiplied the two to estimate LRS. I think a number of issues exist with this approach. First, these means include uncertainty in their calculations that are not being carried forward in the calculation. Second, I think it is very likely given that females experience different selective pressures in different sized units that individual females may invest in reproduction differently throughout their lives and extrapolating reproductive rates this way does not take different reproductive strategies into account (this assumes that reproductive rates are fixed over the life of an animal when age can certainly influence this a lot, for example). Finally, I'm not sure I fully understand the value of making these calculations rather than sticking to the data, or calculating LRS on a subset of females that have been observed over their whole lives.

Author's reply: We agree that the original LRS estimations were coarse and could have better represented our current data. In light of these suggestions, we have made two changes. First, we have modified our estimation by using median survival times at each group (i.e., instead of the fixed mortality rates). Second, we now clarify in L228-232 that these estimates are meant to reflect an "average" female that stays at a given unit size category throughout her life (which, as we mention in L229-230 and L420-445, is not always the case). These estimates are solely meant to highlight the need for reproductive strategies like those suggested; however, more data is needed to identify these strategies and whether they offset the potential fitness costs we've detected here. See line-by-line comments below.

2. My second concern is about some of the language in the manuscript which I found confusing in many places. I pointed out these areas in the comments below, but two recurring instances included (i) "mortality" being used for both adult females, as well as infants, and (ii) "female

success”, “infant success” and “success” being used throughout (perhaps interchangeably?) where I think you simply mean to say infant surviving to weaning.

Author’s reply: We agree that this wording was confusing. We now (i) separate adult and infant mortality as “female death rates” and “infant mortality” respectively and (ii) have replaced all references to “successful offspring” with “surviving offspring.” See line-by-line comments below.

Overall, I enjoyed the manuscript and found the results and story compelling. I include questions and suggested clarifications below. I wish the authors luck in revising this work.

REVIEWER COMMENT	REPLY
Introduction	
Line 41: I think the costs and benefits of group living could be made a bit more general here when you introduce them. Competition when foraging and predation risk are two examples that apply to your study system, but there are many others like competition for or access to mates, communal resource defence, thermoregulation, cooperative breeding/parental care.	We added additional possible costs/benefits to our first paragraph (L42-44).
Line 70: It’s important to distinguish between within and among-group competition when discussing social species. I believe here you mean within-group feeding competition? But in some social species larger groups better defend food resources and thus have reduced competition for food from belonging to larger units.	We have added “within-group” and “between-group” at various places throughout the manuscript (e.g., L55, L61) and have made sure to stick with “within” vs. “between” group (rather than “intra-” vs. “inter-”) terminology throughout.
Line 72: Here, I think it gets confusing because you must have female fitness in mind specifically but I thought this was still a general section on group size and fitness. In your example with infanticide risk, I assume males would have higher reproductive success if they took over a unit with more females, no? Infanticide should increase male fitness but would decrease female fitness. Perhaps discussing conflict in optimal group	We clarified the wording here (L72-74) to reflect that we are focusing on group size effects on females.

size between males and females would help make your point?	
Materials and Methods	
Line 125: how often is each unit seen/observed? A range of lengths of time would be useful for those not familiar with the system.	We typically observe the study units on a near-daily basis. We've now added this wording in L121.
Line 138: If follower males can deter takeovers (and possible infanticide), wouldn't the number of follower males in a group also benefit females? For example, large female units with more follower males can be shielded from takeovers? Is it possible to look at the interaction between female and male group sizes on female mortality or reproductive success?	We controlled for the number of males in all relevant analyses. Contrary to our expectations, males did not impact infant or female survival. We suspect that the beneficial impacts of males may be obscured by their effects on social instability (i.e., we often see many males join units at or just before the time of takeovers). Assessing the impact of males and their participation in unit defense remains a focus for future analyses.
Line 172: Why did you perform the female and infant mortality analyses differently? Why not run survival models for both?	Given the uncertainty around some female age estimates, we initially decided it would be more conservative to estimate death rates while controlling for differences in age (known or estimated, see also our response to R1). In light of this suggestion, however, we now provide the results from Cox mixed effects models of adult female survival, for which we incorporated uncertainty in female ages. These model results are included in the supplementary materials.
Line 180 (also 203, and others): Are these separate random effects, or were they nested in any way?	Given that females can appear in different units in their lifetimes, all random effects were crossed and not nested. We now clarify this in L183-185.
Line 199: "monthly success" is a bit confusing here. I assume you refer to the success of weaning >1 offspring? Perhaps clarify with a more specific term.	We now avoid using "success" throughout the manuscript wherever possible.
Line 204: What is the logic behind using 6-month vs monthly bins? Do geladas have distinct breeding seasons, or can females breed year-round? You could clarify what the	In addition to using 1-month bins (which allow us to more closely track changes in unit size as they occur), the 6-month bins were included as an added check to determine

typical reproductive cycle is within a unit or within one female and how your temporal bins map on to the biology of the system?	whether unit size around the month of conception similarly determined the odds of conceiving a surviving offspring (as opposed to unit size at a fixed time point). Geladas breed year-round but have two distinct conception peaks that map onto these respective bins. We now mention this reasoning in L213-215.
Line 212: Are “successful” offspring the same thing offspring that survived until weaning? I’m not sure if successful is a common term used in primatology research to mean weaned young, but I find it confusing throughout the paper as different researchers may identify “success” with different developmental points.	We agree that this term was confusing and have avoided using it in favor of describing what we actually mean (i.e., an offspring that survived until the average age of weaning for this population).
Line 248: Why would you need to control/offset for total observation time at each unit? Are takeovers so quick or short-lived that units that are less frequently observed are at risk of missing takeover events? My understanding of a takeover is that the dominant male of a unit becomes permanently displaced by a new male, which seems like a rather permanent event. Please clarify whether I am mistaken, or why controlling for total observation time is required.	Takeovers are highly conspicuous events that lead to long-term changes in the male membership of reproductive units. Even in the absence of direct observation, takeovers are inferred whenever a social unit is first observed with a new reproductively dominant male. Offsetting for observation months was purely for statistical purposes. Because the individual data points were continuous periods during which a unit was observed at a given size (# of females), we controlled for observation time (i.e., the number of months the unit was at that size). In other words, if a unit had 4 females for only one month, we were statistically less likely to observe a takeover during that month when compared to a unit of 4 females that maintained that size for 8 months. We’ve changed the wording to clarify this in L264-266.
Results	
Line 276: How does reproductive lifespan differ from lifespan? Are you somehow incorporating reproductive senescence into these estimates, and if so how?	Our previous “reproductive lifespan” estimates were crudely approximated from the mortality rates at each unit size category. In an effort to more directly reflect our current data, we now estimate reproductive lifespans by subtracting the population mean age at

	first birth from the median age at death at each unit size category, accounting for right-censored females (L220-L232). We did not incorporate reproductive senescence into these estimates (as far as we know, females reproduce up until death), but agree that incorporating age effects would provide more robust estimates in future analyses.
Line 285: Are you multiplying the average lifespan of females in different sized units by the average annual reproductive rate across units? This is likely misleading as a measure of lifetime reproductive success as both averages have uncertainty associated with them that is not being carried forward into these calculations. This broad unit-level calculation doesn't incorporate important information about how unit size might shape reproductive decisions or possible competition within a unit. For example, in some social animals variance in reproductive success differs with group size. I would include caveats about this type of calculation up front, at the very least. Otherwise, I'm not sure I trust these numbers and don't understand the purpose of these calculations.	We agree that this is potentially misleading and coarse. In addition to changing our approach to approximating the reproductive lifespan, we now discuss caveats to this calculation in L288-232, mention the additional reproductive parameters that may vary with unit size (L391-393), and highlight the need for more complete lifespans to determine whether and how reproductive strategies vary across units and influence lifetime fitness (L401-403)..
Figure 2A: Can you include post-hoc indications of which group is different from which group rather than the p-value? I cannot tell which means differ from one another based on the figure.	We have now changed Fig 2a and 2b, as the originals were somewhat misleading. The analyses all used unit size as a continuous variable, although we categorized unit size in an effort to match 2c. Now, the plots show the continuous effect of unit size on interbirth intervals.
Discussion	
Line 358: can you clarify whether you mean adult or offspring mortality? It is a bit confusing when you have two mortality variables.	We have clarified this through-out the manuscript: when we refer to adult females, we use "death rates"; when we refer to infants, we use "infant mortality," (e.g., L32; L102).

Line 365: I'm not sure I agree with your projections based on unit-level means multiplied by one another. I would argue that females may be making different reproductive decisions in different sized units based on the pressures they face, particularly if they face different risks of infant mortality as your results show. Without more information on reproductive investment across different sized units I'm not convinced the broad estimates hold true when you extrapolate to lifetime levels. You must have lifetime data for some individuals – in this subset of your data, can you compare lifetime reproductive success across units of different sizes to support or replace these calculations?	Unfortunately, given the long lifespan of female geladas compared to the number of years of our long-term project, reporting the total reproductive output of deceased females of known age (N=8) or known first births (N=31) right now would bias our sample towards short-lived females. Addressing lifetime reproductive success using complete data is certainly a goal for future analyses once more known-aged females have completed their lifespans. The estimate here is meant to help describe the effect of variation in fitness across different groups, with the assumption that female life history strategies, etc., remain constant (which, of course, we know they do not). We have made this clear in our methods and discussion (L228-232, L391-393, L401-403)) and intend to investigate female behavioral responses to unit size, etc., in a future analysis.
Line 377: "infanticide" typo	This has been fixed (L386).
Supplemental tables: Please report random effects	These have been added throughout the supplemental materials.

Appendix B

04 May 2021

Dear Dr. Dall,

Thank you again for taking the time to review our manuscript (RSPB-2021-0313) for publication in *Proceedings of the Royal Society B: Biological Sciences*. We have made the two requested changes from Reviewer 1 (and provide a detailed response below) and are pleased to submit our revised manuscript entitled “The Goldilocks Effect: Female geladas in mid-sized groups have higher fitness.”

All co-authors have read through these comments, the reply to the referees, and the revised manuscript. We do not have any manuscripts in press, under consideration, or published elsewhere that report the findings presented here. We have submitted the manuscript as a preprint to bioRxiv.org (<https://doi.org/10.1101/348383>). All work was conducted in accordance with institutional guidelines at the University of Michigan and the Ethiopian Wildlife Conservation Authority. Data have been deposited on github and reviewers can access the data here: <https://doi.org/10.5281/zenodo.4737602>. There will be no restrictions on data availability after publication.

Sincerely,

Elizabeth Tinsley Johnson and Jacob Feder
(on behalf of all co-authors)

Associate Editor

Board Member

Comments to Author:

The reviewers and I agree that you have addressed all of the comments and the manuscript will make a very nice contribution to the field. There are only two very minor clarifications to fix- one wording clarification and fixing the file in the GitHub repository.

Reviewer(s)' Comments to Author:

Referee: 1

Comments to the Author(s).

The authors have done a great job in addressing my comments. The revised version provides helpful additional context and points out how future research might built on the described findings. I only have two minor comments:

In the revised version you report that the death rate of adult individuals in small groups is 10.4% and in large groups 7.3% - but in both small and large groups you estimate that individuals have median age at death of 16 years (line 292). So, for a given age, is there higher mortality in large groups, but because there are more young individuals at start of group in the larger group, the expected lifespan is the same?

REPLY: We clarified the way we present this analysis, which was simply used to estimate median reproductive lifespans by unit size. Therefore, we moved the calculated median ages at death result to the “Reproductive Performance” results section (L299-302) and we clarified how we calculated the median ages at death by unit size from survival curves in the caption for supplemental figure S3. Specifically, median ages at death were determined from where the survival curves crossed the 0.5 threshold of adults surviving (now indicated by a dotted horizontal line in the figure). Although annual death rates differ between small and large units, median ages at death are roughly equivalent, perhaps as a result of modest differences in the age distributions of these groups.

- It appears that during the latest update of the GitHub repository the data file was deleted (there are now two code scripts, one apparently the original another revised). I went to the earlier version of the repository and could download the file, so everything seems to be there. I think it would be good to update this. My suggestion would also be to deposit the data at a repository with a DOI so that it is more permanently available (avoiding the risk that a collaborator closes their GitHub repository). I can recommend KNB, which is free and provides helpful settings to add metadata for ecological datasets (which increases discoverability) and add a licence (which clarifies and increases reusability): <https://knb.ecoinformatics.org>

The code does refer to an additional input data file that does not seem to be part of the rdata package - the file cyphers_dobs.csv which appears to be used to assign dates of births to females who were alive at the beginning of the study. Is this preprocessing and

the resulting output is part of the rdata package?

REPLY: Thank you letting us know! We've fixed the issue with the file on GitHub and set up a DOI for the repository (<https://doi.org/10.5281/zenodo.4737602>).

Referee: 2

Comments to the Author(s).

I previously reviewed this manuscript, and I thank the authors for their work in revising and explaining their research in the resubmission. I think the manuscript is generally much clearer and represents important work. Nicely done.